# Research on the Characteristics and Influencing Factors of Provincial Urban Network from the Perspective of Local Governance—Based on the Data of the Top 100 Enterprises in Four Categories in Fujian Province

**Jialiang Zhao [1,2], Suqiong Wei [1] and Qingmu Su [2,*]**

[1] School of Geographical Sciences, Fujian Normal University, Fuzhou 350007, China; zhaojl83@126.com (J.Z.)
[2] School of Architecture and Planning, Fujian University of Technology, Fuzhou 350118, China
[*] Correspondence: martain@foxmail.com

**Abstract:** With the development of the division of labor in product value chains and the specialization of urban functions, the network link structure model among cities is being reshaped. Studying the structure of urban networks and its related theories in the context of scale, place and policy is still an open area. This study constructs a research framework to study the urban network formed by the synergy of scale, place and policy. It mainly takes enterprises in different industries in different provinces as the empirical scale and object, and uses methods such as a social network and Geo Detector to analyze the characteristics and influencing factors of the provincial network relationship mode of enterprises among cities. The main findings are as follows. (1) Firstly, the urban network linkage in general shows strong coastal centrality and small-world network characteristics. The urban network linkages reflected by different types of enterprises all have obvious spatial directionality and polarization effects. (2) Coastal cities have strong centrality, and the specialized division of urban functions emerges, with large cities becoming a concentration area for different types of corporate headquarters, while small- and medium-sized cities carry a large number of processing and assembly enterprises. (3) The networks of different types of enterprises have different sensitivity to each influencing factor. For example, emerging industries have the strongest correlation with the economic scale and social services; manufacturing industries are most influenced by the public services, administrative level and development zone level; and service industries are most influenced by science and technology expenditure and the same metropolitan area. In conclusion, this study contributes to the understanding of network heterogeneity at the provincial scale and provides policy support for the local governance scale, as well as promotes the expansion of the urban network theory to network governance applications in the "flow space".

**Keywords:** local governance; urban network characteristics; synergy framework; provincial scale; enterprise network perspective

## 1. Introduction

In the context of globalization in the 21st century, as ICT and transport technologies continue to innovate, production fragmentation has profoundly changed the global economic landscape and led to the emergence of urban networks based on the division of labor in product value chains and the specialization of urban functions, which therefore has reshaped the network link structure model [1–3]. The world is in a high-speed "flow space", a phenomenon that leads to the fact that factors are not only clustered in a close geographical space, but can also interact with each other between cities to generate externalities thanks to dense intercity linkages [4,5]. The multi-scale nested urban networks formed by such competitive and cooperative relationships have become an important platform for regional economic development [6]. In the context of the 'relational turn' and 'flow

space' in contemporary economic geography research, it has become the mainstream way to study the relationship between cities from the perspective of enterprise networks. Urban networks in China are multi-scale and complex, covering a large geographical area with significant regional differences. Under the role of different geographical environments and the coordination of scale, place and policy [7], the patterns of urban network relationships and influencing factors become diverse [8]. As a result, there is still a need for a breakthrough in the analysis of urban networks and the refinement of related theories in the context of scale, place and policy.

Research from an enterprise network perspective facilitates the quantification of local-scale network frameworks. Driven by Sassen's advanced producer services (APS) [9] and the enterprise linkage research ideas of the "Globalization and World Cities Research Group (GaWC)" [10], research on urban networks based on enterprise networks has been conducted from four main aspects. First, in terms of research perspectives, urban network characteristics [11,12], urban network pattern evolution [13,14], urban network externalities [15] and urban network resilience [16] have become important tools for analyzing urban networks at different scales [17]. Secondly, in terms of formation mechanisms, the structural characteristics and evolution of urban networks have been explored from the perspective of frameworks, such as the externality interrelationship framework [18], the resilience framework [19], the dynamics framework [20], the economic mechanism framework [21] and the 'behavior–structure–performance' framework [22]. Thirdly, in terms of the spatial scale of research, macroscopic scales involving global, national and regional scales remain the main object of research [23], and research at provincial and sub-provincial scales is still lacking. Thus, there is a lack of implementable local spatial governance frameworks. As the basic unit of a province, a city is the basic component of analyzing the internal network of a province. By quantifying the network relationship of each city in this study, the network relationship between cities can be grasped from the provincial level as a whole. Fourthly, in terms of research objects and methods, new data sources of enterprise dynamics have been continuously expanded, with financial industry, advanced productive services [9] and large-scale multi-location enterprises with different properties as the objects [24–27], which are the mainstream of research in recent years. However, due to the heterogeneity of industries, the research conclusions are also different, so there is still room for further deepening the study of urban networks from the industry perspective. This study explores the urban network relationship of enterprises from the perspective of the corporate network, and can quantify its urban network mechanism from specific things. To sum up, how to study the formation mechanism of urban networks under the coordination of scale, place and policy from the perspective of different industries is a concrete framework to realize the enforceability of local space governance.

In past studies, the main research methods for urban network characteristics include the chain network model [28], compromise network model [12] and headquarter–branch affiliation model [9]; the main research methods for urban network connections are the Social Network Analysis [29] and other research methods; the methods used to identify the influencing factors of urban networks are the Ordinal Logit Model, Multinomial Logit Model [22] and Quadratic Assignment Procedure (QAP) [11]. In order to quantify the structure of urban networks under the coordination of scale, place and policy, this study draws on the strengths of various methods and uses the Headquarter–Branch Affiliation Model and Social Network Analysis Methods to analyze the urban affiliation structure of different types of enterprises, as well as the Geo Detector to explore the impact of different influencing factors on the spatial layout of enterprises. This is a response to how governance is dynamically configured at the provincial scale.

As we can see above, the research on urban networks based on enterprise networks has yielded rich results, but at the same time, the academic community has become increasingly aware of the fact that the research on existing urban networks has neglected the influence of the industry development characteristics and institutional environment [26,30], ignored the research on the local regional scale such as provincial areas [31], lacked dialogue with

local governance policies and involved insufficient policy evaluation. Therefore, this paper attempts to address the above issues by integrating the post-disciplinary perspectives of relational economic geography, evolutionary economic geography and political economic geography as well as current spatial governance theories. Using the research framework of the coordination of scale, place and policy, this paper takes enterprises in different industries in Fujian Province as the empirical scale and object to explore the characteristics and influencing factors of the provincial urban network patterns corresponding to different enterprise networks. The main objectives are, firstly, to establish a research framework of urban network relations under the coordination of scale, place and policy, which can be used to explore the networked characteristics and governance issues of provincial urban development under the flow space, and secondly, to explore the relationship between different types of enterprises and the influencing factors, and to regroup local and non-local assets for different types of enterprises to take advantage of key resources and form a specialized urban functional division of labor.

## 2. Methodology

### 2.1. Research Cases and Data Sources

This study takes Fujian Province as a case study, and the main starting points are (1) Fujian Province is located on the southeast coast of China, and now governs nine cities and one district (Pingtan Comprehensive Experimental Zone). The geographical units of mountains and seas in Fujian Province differ significantly. It was an important pilot area for the advancement of industrialization and market-oriented reforms in early China. (2) The focus of the province's regional development policy has gone through a process from the coastal Xiamen–Zhangzhou–Quanzhou region to the Fuzhou region and to the western mountainous regions. At present, metropolitan circles of different sizes and administrative levels have been formed. (3) At the same time, Fujian Province is a province with frequent interactions between central and local levels of urban power, and the policy governance process is continuous, typical and representative. (4) There are more than 1.32 million enterprises with a registered capital of more than 1 million, which have a data foundation for a network analysis.

The data for this study were obtained from the list of Top 100 Enterprises in Fujian Province in 2021, jointly published by the Fujian Enterprises and Entrepreneurs Confederation (FJEEC), Fujian Media Group and Fujian Academy of Social Sciences, with a total of 307 enterprises, covering 3 major industries, namely the List of Top 100 Manufacturing Enterprises in Fujian Province, the List of Top 100 Service Enterprises in Fujian Province and the List of Top 100 Emerging Enterprises in Fujian Province. The manufacturing industry in this study refers to the industry in the national economy that uses certain resources (materials, energy, equipment, etc.) to transform them into products that can be used and utilized by people through the process of mechanical manufacturing. The service industry refers to the collection of production departments and enterprises engaged in service products. Emerging industries refer to industries that are based on major technological breakthroughs and major development needs, have a major leading role in the overall economic and social development and long-term development and have huge growth potential. In this study, Qixinbao's enterprise association genealogy panel was used to collect data related to the corporate registered addresses and enterprise types of the headquarters and branches of the top 100 enterprises in Fujian Province in 2021, and finally 15,571 branches were screened out from 46,086 branches through data cleaning and processing (excluding enterprises with registered locations outside Fujian Province and enterprises with a non-existing status such as cancelled or revoked). After that, the relevant data matrix was formed through the correlation between the headquarters and branches of enterprises. The distribution of the headquarters of the top 100 enterprises and the distribution of specific industries are shown in Figure 1.

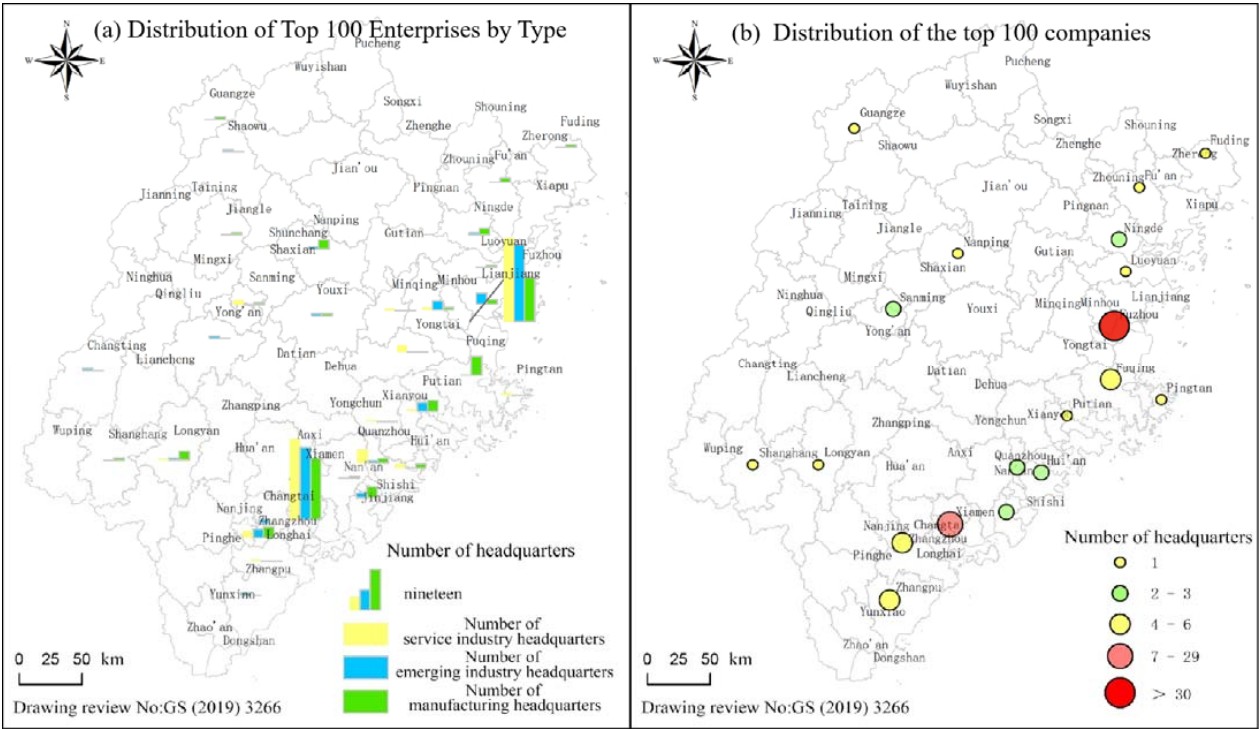

**Figure 1.** Spatial distribution of headquarters of the top 100 enterprises in Fujian Province.

*2.2. Research Framework*

This study focuses on urban network relationships in the context of scale, place and policy synergies in urban network governance, and is an extension of the 'flow space' description of urban networks to network governance applications. The research framework of this study can therefore be divided into two parts (Figure 2). The first part is the construction of urban network relationships. Firstly, this study applied the data of the top 100 enterprises to build a matrix of urban linkages between the headquarters and branches of enterprises through the Matrix Construction Method. Further, Social Network Analysis Methods (network identity, network centralization and centrality) are applied to analyze the urban link structure of different types of enterprises, including manufacturing, emerging industries and service industries. The analysis focuses on the network connection, network status and network link mode, which provide the dependent variables for the next stage of the network factor analysis. The second part of the analysis is based on the impact of the scale, place and policy detector factor on urban networks. Firstly, the main influencing factors of scale, place and policy are developed. The Geo Detector is then used to explore the impact of different influencing factors on the spatial layout of enterprises. This helps us to establish a governance scale for the dynamic allocation of different types of enterprises and provides scientific evidence for the formulation of provincial development policies.

*2.3. Research Method*

(1)    Matrix Construction Method

The linkage intensity between city a and city b, C (a,b), reflects the strength of the network linkage established between the two cities through all corporate headquarter–branch linkages and is calculated as follows:

$$C(a, b) = \sum_{i=1}^{m} v_{ia} v_{ib} \tag{1}$$

where m denotes the number of enterprises with headquarters or branches in both city a and city b; $v_{ia}$ and $v_{ib}$ denote the number of enterprises i in city a and city b, respectively.

Based on the above ideas, this study constructed a matrix of urban links in Fujian Province with the help of a computer program.

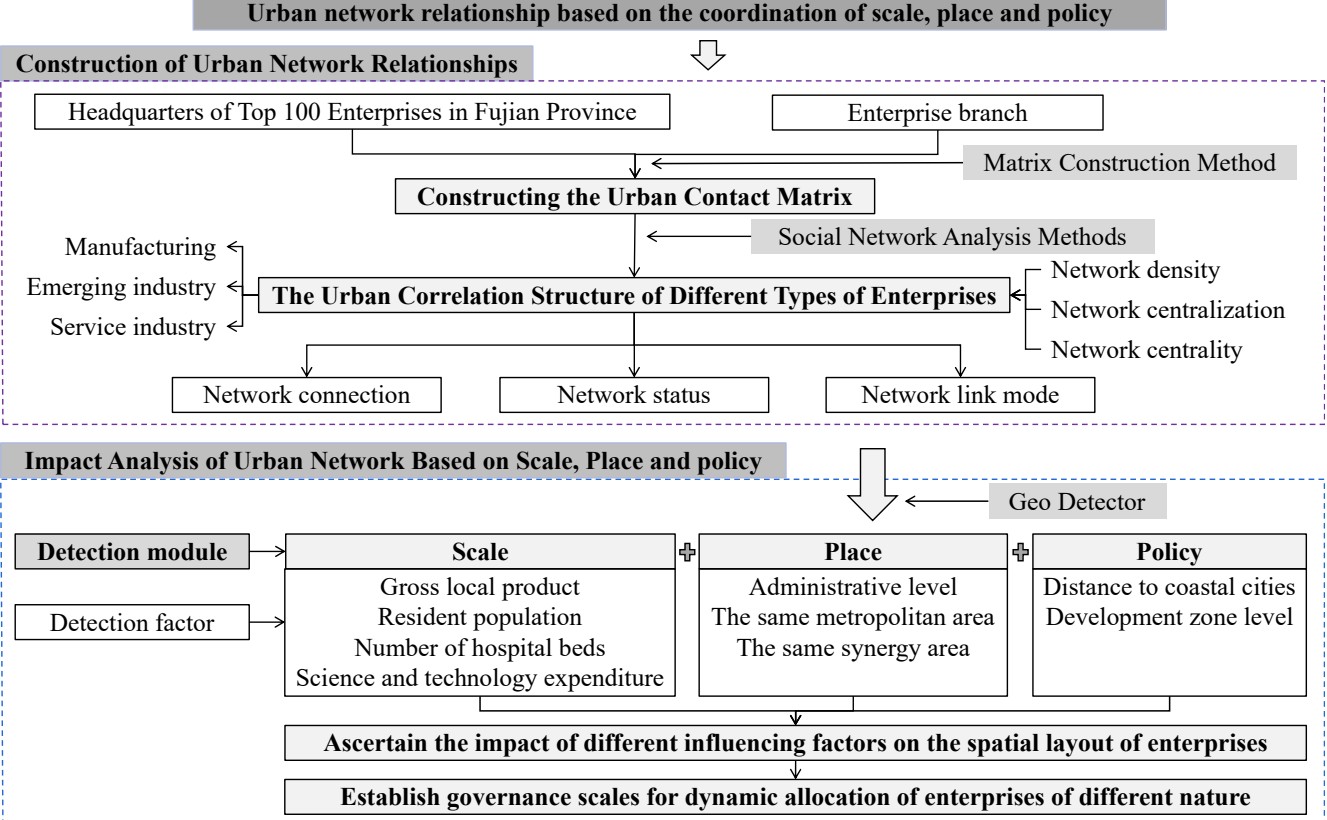

**Figure 2.** Research framework for urban network relationships based on the coordination of scale, place and policy.

(2)    Social Network Analysis Methods

A Social Network Analysis (SNA) is an analytical method to describe the overall shape, characteristics and structure of a network, and its core idea is to study the related structural issues based on the "relationship" perspective. In this paper, we mainly use Social Network Analysis Methods to explore the characteristics of network density, network centralization, centrality, etc. of the urban link structure in Fujian Province, and then we can obtain the network connection and network status of the overall characteristics of the urban network. These methods are relatively mature, and the details are shown in the studies by Rong Shengke and others [11,29]. Meanwhile, the network centrality of each city can be used as the dependent variable for analyzing the influence factors. Network density refers to the closeness of the relationship between nodes in the network, which expresses the closeness of the connection between nodes. Centrality reflects the status of nodes in the network. Network centralization is an analysis of network integration based on centrality, which shows the overall integration of the network.

Calculation Formula (2) of network centrality $C_{deg}(v)$:

$$C_{deg}(v) = \frac{d_v}{|N|-1} \tag{2}$$

Calculation Formula (3) of network centralization $C_D$:

$$C_D = \frac{\sum_{v \in N} \max\left(C_{deg}\right) - C_{deg}(v)}{|N|-2} \tag{3}$$

Calculation Formula (4) of network density D:

$$D = \frac{2L}{N(N-1)} \tag{4}$$

where $d_v$ is the number of neighbors of the node, N is the set of all nodes in the network, $|N|$ is the number of nodes in the network, $\max\left(C_{deg}\right)$ is the maximum value of the network centrality and L is the connection possessed by the network practice.

(3)　City Rank Score

The city rank score Si of a county or city is equal to the product of the number and weight of corporate headquarters established in that city, plus the product of the number and weight of branches established in that city, as shown in Formula (5).

$$S_i = Q_{ih} \times q_h + Q_{ib} \times q_b \tag{5}$$

where $Q_{ih}$ refers to the number of headquarters of top 100 enterprises in city *i* (*h* for headquarters); $Q_{ib}$ refers to the number of branches in that city (*b* for branches); and *q* is the weight ($q_h = 2$, $q_b = 1$).

This study assigned scores of 2 and 1 to the headquarters and branches of enterprises, respectively. According to Formula (5), the scores of counties and cities in the urban network of Fujian Province are calculated and classified into five levels according to the natural fracture method, and thus the network hierarchy and multidimensional nested schema of cities can be explored.

(4)　Geo Detector

The Geo Detector is a set of statistical methods for detecting spatial variability and revealing the driving forces behind it. Its core idea is based on the assumption that if an independent variable has an important influence on a dependent variable, then the spatial distribution of the independent variable and the dependent variable should be similar. This paper uses the Factor Detection module in the Geo Detector to study the magnitude and variation in the degree of explanation of each influencing factor in the centrality of urban enterprise networks in Fujian. The expressions of the Factor Detection module are as follows:

$$q = 1 - \frac{\sum_{h=1}^{L} \sigma_h^2 N_h}{N \sigma^2} \tag{6}$$

where q is the value of the detection power of the influencing factor on the network centrality; h = 1, . . . ; L is the grading of each factor of the variable; $\sigma^2$ is the total variance of the network centrality of cities in Fujian Province, and $\sigma_h^2$ is the variance of the network centrality of cities in Fujian Province; N is the number of cities in Fujian Province; and Nh is the number of types of the influencing factor X. The value range of q is [0, 1], and the larger the value q, the greater the influence of the factor on Fujian.

### 2.4. Establishment of Influencing Factor Indicators for Urban Networks

The scale, place and policy synergy framework perspective proposed by Gong [7] provides an integrated analytical thinking of scale, place and policy synergy for understanding the evolution of regional economic development. It deepens the understanding of the political–economic spatial unity of economic network linkages and political governance spatial reconfiguration, enables the analysis of the regional economy "beyond borders" [1] and promotes the development of the network paradigm-based relational geography theory to network governance applications. Scale refers to one or several levels of representation, experience and organization of geographic events and processes. In this research, this study refers to the administrative scale of provinces, cities and counties, and the governance scale of metropolitan areas. Place refers to a specific area relative to large-scale areas such as the world and a country, and is a city or county area with special natural and socio-economic

factors. Policy refers to government-led development planning, spatial development priorities, the layout of special economic zones and other specific socio-economic measures. Among the local factors, the city population size and economic scale influence the linkage relationship between cities, and the contribution of different industries in the economic network varies [25]. The level of local public facilities, science and technology expenditure and social service conditions also has an important impact on the layout and development of enterprises [32,33], and these key resources become important factors to be considered when selecting the location of enterprise networks. Therefore, this study used the indicators of the gross local product, resident population, number of hospital beds and science and technology expenditure to quantify the local influencing factors. At the scale level, the preferences of firms' location choices and the network concentration process they lead to are important micro mechanisms of urban relationship patterns [21,34]. Governments of different administrative levels have different resource disposal capabilities. The higher the administrative level, the more political power, policy advantage and information advantage it has, which is conducive to enhancing the competitiveness of the product value chain and directly affects the location choice of company headquarters and branches [34]. The metropolitan area development and synergy area development promoted by the state are conducive to the formation of enterprise networks near the region [31]. Therefore, in this study, the administrative level, the same metropolitan area and the same synergy area indicators are chosen to quantify the influencing factors of the scale. At the policy level, the policy of coastal priority development is conducive to enterprise network development [35]. Fujian Province implements the development policy of coastal cities driving inland cities. Special development zones established by the state build economic links between different spatial scales, and it is conducive to attracting business clusters. The spatial organization of multi-zone enterprises is influenced by the regional development context, development regime and policies [1,2,31,35]. Therefore, two indicators, the distance to coastal cities and development zone level, were chosen in this study to quantify the influencing factors of policies.

In this study, nine indicators (Table 1) are used as independent variables, and the network centrality of each city is selected as the dependent variable to explore the relationship between the sensitivity of different types of enterprise networks and each factor through the Geo Detector. Among them, the administrative level indicator refers to the weights of 5, 3, 2 and 1 assigned to provincial capital cities (special zones), prefecture-level cities, county-level cities (districts) and counties by each city according to its attributes. The distance to coastal cities indicator refers to the geographic distance from a city center to a coastal city center calculated using arcGIS.

**Table 1.** Indicators of influencing factors on urban networks of top 100 enterprises.

| Detection Module | Sequence | Detection Factor |
|---|---|---|
| Local economic scale | X1 | Gross local product |
| | X2 | Resident population |
| Local key resources | X3 | Number of hospital beds |
| | X4 | Science and technology expenditure |
| Scale | X5 | Administrative level |
| | X6 | The same metropolitan area |
| | X7 | The same synergy area |
| Policy | X8 | Distance to coastal cities |
| | X9 | Development zone level |

## 3. Result Analysis

*3.1. Analysis of the Overall Characteristics and Differences of Urban Networks*

3.1.1. Network Linkage: Strong Coastal Centricity and Small-World Network Characteristics

Using the social network method to measure the related indicators of the urban network and according to Formulas (1)–(4), we find that the results of the top 100 enterprises and three major industries showed overall poor connectivity, strong centripetal concentration, incomplete development and small-world characteristics (The main feature of a small-world network is that it has a relatively small average path length and a relatively large clustering coefficient; that is, the clustering of the three major industries is high, but the systems are relatively independent). The overall urban network includes 64 cities, with a wide network coverage. The single-valued network density (a network formed by binarizing the network connection strength according to the standard of the network connection strength) is 0.07, and the network centralization is 0.79, indicating that the direct connectivity between network nodes is low and has typical centripetal characteristics. The multivalued network centralization is 0.08, indicating that in addition to the obvious centripetal characteristics of the urban network, the connection between the core city and the network center city is weak. From the analysis of the three major industries, it can be found that the single-valued network densities of the service, manufacturing and emerging industries are 0.066, 0.065 and 0.069, respectively, with the single-valued network centralization of 0.77, 0.70 and 0.57, respectively, while the multivalued network centralization is 0.06, 0.07 and 0.08, respectively, indicating that network connectivity of the three industries is similar to the total top 100 enterprises, and the centripetal characteristics of emerging industries are weaker compared to other industries.

In the urban network, most nodes only have a small number of connections with other nodes, and the small-world feature is obvious. Take the top 100 enterprises as an example (Table 2). There are 262 city pairs in the whole city network. The higher the connection intensity, the fewer the city pairs. Only 3.44% of them have a connection intensity of more than 100, and 7.63% have a connection intensity of 50–100 and above. Among them, Fuzhou–Xiamen, Fuzhou–Pingtan and Zhangzhou–Xiamen have the first level of connection intensity (above 186), and Fuzhou–Xiamen has the largest connection intensity (460); Xiamen–Fuzhou, Fuzhou–Zhangzhou, Fuzhou–Minhou, Fuzhou–Fuqing, Xiamen–Zhangzhou, Fuzhou–Longyan, Fuzhou–Putian, Fuzhou–Sanming, Zhangzhou–Zhangpu and Zhangzhou–Fuzhou have the second level of connection intensity, ranging from 83 to 186; the connection intensity of the other cities is weak (below 83). The pattern of linkages between manufacturing, service and emerging industries also exhibits this hierarchical imbalance, with significant spatial directionality and polarization effects. This indicates that on the one hand, enterprises still tend to establish branches in cities with similar geographical and geographical locations, leading to regional characteristics in urban chain relations. On the other hand, the strength of links between coastal cities has greatly improved, with high-level reciprocal links and preferential links among high-level cities being the dominant ones. On the other hand, the intensity of links between coastal cities has increased substantially. High-ranking cities are dominated by high reciprocal links and merit-based links among themselves. At the same time, the influence of coastal central cities has transcended administrative boundaries. The coastal central cities have incorporated other cities in the long distance of mountainous areas into their own hinterland areas, and the network hinterlands have begun to overlap with each other with blurred spatial boundaries.

**Table 2.** Top 10 counties of network in Fujian Province and net outflow.

| Top 100 Enterprises | | Manufacturing | | Emerging Industry | | Service Industry | |
|---|---|---|---|---|---|---|---|
| Contact Direction \| City | Net Out-Degree | Contact Direction \| City | Net Out-Degree | Contact Direction \| City | Net Out-Degree | Contact Direction \| City | Net Out-Degree |
| Fuzhou→Xiamen | 460 | Fuzhou→Xiamen | 147 | Xiamen→Fuzhou | 31 | Fuzhou→Xiamen | 588 |
| Fuzhou→Pingtan County | 263 | Zhangzhou→Xiamen | 86 | Fuzhou→Fuqing | 17 | Xiamen→Fuzhou | 277 |
| Zhangzhou→Xiamen | 237 | Fuzhou→Minhou County | 69 | Zhangzhou→Xiamen | 14 | Zhangzhou→Xiamen | 234 |
| Xiamen→Fuzhou | 186 | Quanzhou→Putian | 36 | Fuzhou→Pingtan County | 11 | Fuzhou→Pingtan County | 199 |
| Fuzhou→Zhangzhou | 154 | Xiamen City→Minhou County | 34 | Quanzhou→Fuzhou | 10 | Xiamen→Zhangzhou | 173 |
| Fuzhou→Minhou County | 140 | Xiamen→Fuzhou | 30 | Xiamen City→Zhangpu County | 9 | Fuzhou→Zhangzhou | 163 |
| Fuzhou→Fuqing | 122 | Fuzhou→Longyan | 29 | Fuzhou→Minhou County | 8 | Fuzhou→Sanming | 111 |
| Xiamen→Zhangzhou | 110 | Fuzhou→Pingtan County | 28 | Fuzhou→Nanping | 7 | Fuzhou→Minhou County | 101 |
| Fuzhou→Longyan | 109 | Fuzhou→Nanping | 23 | Xiamen→Putian | 7 | Quanzhou→Xiamen | 93 |
| Fuzhou→Putian | 96 | Xiamen→Zhangzhou | 23 | Fuzhou→Huian County | 6 | Fuzhou→Fuqing | 89 |

There are differences in the direction and intensity of urban network linkages reflected by different types of enterprise networks. From the linkage intensity and net out-degree values in Table 1, it can be seen that service industry > overall > manufacturing industry > emerging industry, indicating that enterprises in the service industry are multi-location enterprises and generally have an extensive linkage, while the emerging industry is in the early stage of development, with a multi-centralized layout. The top 10 cities with high linkage intensity generally exist as the cities with a higher administrative ranking, and only a few coastal counties such as Minhou County, Fuqing City, Pingtan County, Zhangpu County and Hui'an County are in the ranks of high contact direction.

3.1.2. Network Status: Strong Coastal City Centrality, Specialized Division of Urban Functions

The weighted degree, out-degree and in-degree indicators calculated for the four sets of enterprise network data reflect the urban network centrality and network status, all with obvious administrative centrality and coastal centrality (Table 3). Firstly, the top 10 cities in terms of city network centrality are all coastal cities and counties except Longyan and Sanming, which reflects the preference of coastal locations by enterprises. This results in strong coastal urban network centrality, making coastal cities the core competitiveness of the urban network in Fujian Province. Secondly, except for Nanping, which is not in the top 10 urban network centers, all other prefecture-level cities and the Pingtan Comprehensive Test Zone have a high network status and centrality, indicating that administrative rank is the key to the location of enterprises.

The network data of four groups of enterprises all show that the out-degree of the urban network is high, but the in-degree is not necessarily high. For example, in manufacturing and service industries, the out-degree and in-degree sequences of Fuzhou and Xiamen are not consistent and differ significantly, indicating that the agglomeration and outward radiation control ability of headquarters and branches in cities are not exactly positively correlated, which is different from the findings of some scholars [34].

**Table 3.** Top 10 cities in Fujian Province in terms of urban network centrality.

| | City | Weighted Degree | City | Standardized Out-Degree | City | Standardized In-Degree | | City | Weighted Degree | City | Standardized Out-Degree | City | Standardized In-Degree |
|---|---|---|---|---|---|---|---|---|---|---|---|---|---|
| Top 100 Enterprises | Fuzhou | 2595 | Fuzhou | 1.00 | Xiamen | 1.00 | Service Industry | Fuzhou | 2530 | Fuzhou | 1.00 | Xiamen | 1.00 |
| | Xiamen | 1632 | Xiamen | 0.36 | Fuzhou | 0.44 | | Xiamen | 1933 | Xiamen | 0.46 | Fuzhou | 0.47 |
| | Zhangzhou | 904 | Zhangzhou | 0.27 | Pingtan | 0.38 | | Zhangzhou | 888 | Zhangzhou | 0.25 | Zhangzhou | 0.38 |
| | Quanzhou | 580 | Quanzhou | 0.18 | Zhangzhou | 0.36 | | Quanzhou | 589 | Quanzhou | 0.20 | Pingtan | 0.29 |
| | Pingtan | 311 | Ningde | 0.03 | Zhangpu | 0.26 | | Pingtan | 281 | Longyan | 0.03 | Quanzhou | 0.18 |
| | Zhangpu | 217 | Sanming | 0.02 | Min Hou | 0.25 | | Sanming | 202 | Ningde | 0.03 | Sanming | 0.16 |
| | Longyan | 208 | Longyan | 0.02 | Quanzhou | 0.22 | | Longyan | 174 | Sanming | 0.02 | Putian | 0.14 |
| | Min Hou | 203 | Putian | 0.01 | Longyan | 0.21 | | Putian | 161 | Putian | 0.01 | Min Hou | 0.14 |
| | Ningde | 179 | Nanping | 0.01 | Putian | 0.20 | | Ningde | 158 | The remaining | | | |
| Manufacturing | Fuzhou | 643 | Fuzhou | 1.00 | Xiamen | 1.00 | Emerging Industry | Fuzhou | 157 | Fuzhou | 1.00 | Fuzhou | 1.00 |
| | Xiamen | 494 | Xiamen | 0.42 | Min Hou | 0.45 | | Xiamen | 118 | Xiamen | 0.88 | Xiamen | 0.50 |
| | Zhangzhou | 227 | Zhangzhou | 0.30 | Fuzhou | 0.30 | | Quanzhou | 54 | Quanzhou | 0.36 | Putian | 0.43 |
| | Quanzhou | 148 | Quanzhou | 0.20 | Longyan | 0.25 | | Zhangzhou | 46 | Zhangzhou | 0.28 | Ningde | 0.33 |
| | Min Hou | 116 | Longyan | 0.05 | Zhangpu | 0.22 | | Putian | 25 | Ningde | 0.06 | Quanzhou | 0.31 |
| | Longyan | 90 | Ningde | 0.04 | Zhangzhou | 0.21 | | Ningde | 24 | Longyan | 0.06 | Zhangzhou | 0.31 |
| | Zhangpu | 59 | Sanming | 0.02 | Putian | 0.21 | | Longyan | 19 | Putian | 0.02 | Fuqing | 0.31 |
| | Putian | 56 | Fuqing | 0.01 | Pingtan | 0.16 | | Fuqing | 18 | The remaining | 0.01 | Pingtan | 0.30 |
| | Ningde | 43 | Nanping | 0.01 | Nanping | 0.14 | | Pingtan | 17 | | | Longyan | 0.24 |
| | Pingtan | 43 | Shanghang | 0.01 | Quanzhou | 0.13 | | Zhangpu | 13 | | | Zhangpu | 0.22 |

The network data of four groups of enterprises reveal the order of the urban network status. Urban specialization formed along the functional division of the value chain is a distinctive feature of the economic landscape of China's urban system. With the deepening of the spatial division of labor in the product value chain, the specialization of city functions is becoming more and more common. Corporate headquarters are gradually separated from R&D and production organizations and clustered in cities of different classes according to their respective locational needs, bringing about differences in the order of the status of different cities in the network system formed by the spatial organization of enterprises. In the urban system, large cities such as Fuzhou, Xiamen, Zhangzhou and Quanzhou have become agglomerations of company headquarters, while small- and medium-sized cities such as Zhangpu, Fuqing and Minhou host a large number of processing and assembly enterprises. New economic linkages are established between cities based on intra-product division of labor [36]. Meanwhile, with the emergence of Ningde, Pingtan and Zhangpu as regional "novelty" cities, a new path of regional development is created with the formation of a specialized division of labor based on services, emerging industries and manufacturing.

### 3.1.3. Network Linkage Patterns: Significant Differences in Network Hierarchy and Multidimensional Nested Schema

There are significant differences in the hierarchical structure and network linkage relationship patterns of the four enterprise factor flow networks (Figure 3). Taking the urban network of the total top 100 enterprise networks as an example, the ratio of each tier is 3:2:6:8:45, among which Zhangzhou, Fuzhou and Xiamen are the first-tier cities with scores greater than 500; Pingtan and Quanzhou are the second tier with scores of 301–500; and developed coastal counties such as Zhangpu, Minhou, Fuqing, Putian and the inland prefecture-level cities Longyan and Sanming are the third tier with scores between 151 and 300. The rest of the cities have scores less than 150, which are the fourth and fifth tiers, and such cities are mainly located in the less economically developed western mountainous areas of Fujian Province. Overall, it presents a hierarchical structure with multiple centers, flattening and ordering, and the link relationship presents characteristics such as a highly dense coastal urban network, a mountain sea linkage axis and two coastal metropolitan areas, namely, a multiple intensive nested schema with a coastal core circle layer, coastal and mountain sea axes and a networked structure. The ratios of cities in the service and manufacturing sectors are 1:1:2:10:50 and 1:1:4:10:48, respectively, showing a hierarchical pyramidal city hierarchy with Xiamen as the monocenter, and a multidimensional nested schema of the coastal core, coastal axis and networked structure. The hinterland of the manufacturing linkage is more extensive, with stronger mountain and sea connectivity. Although Fuzhou has a smaller functional score than Xiamen, the network hinterland is more extensive. The urban network formed by emerging industries forms a dual-center pattern of Xiamen and Fuzhou. The ratio of each level in the urban network is 2:4:7:8:43, and coastal prefecture-level cities and developed counties and cities are at the second level, showing a coastal-belt type and a radial link relationship model with Xiamen and Fuzhou as the center circle.

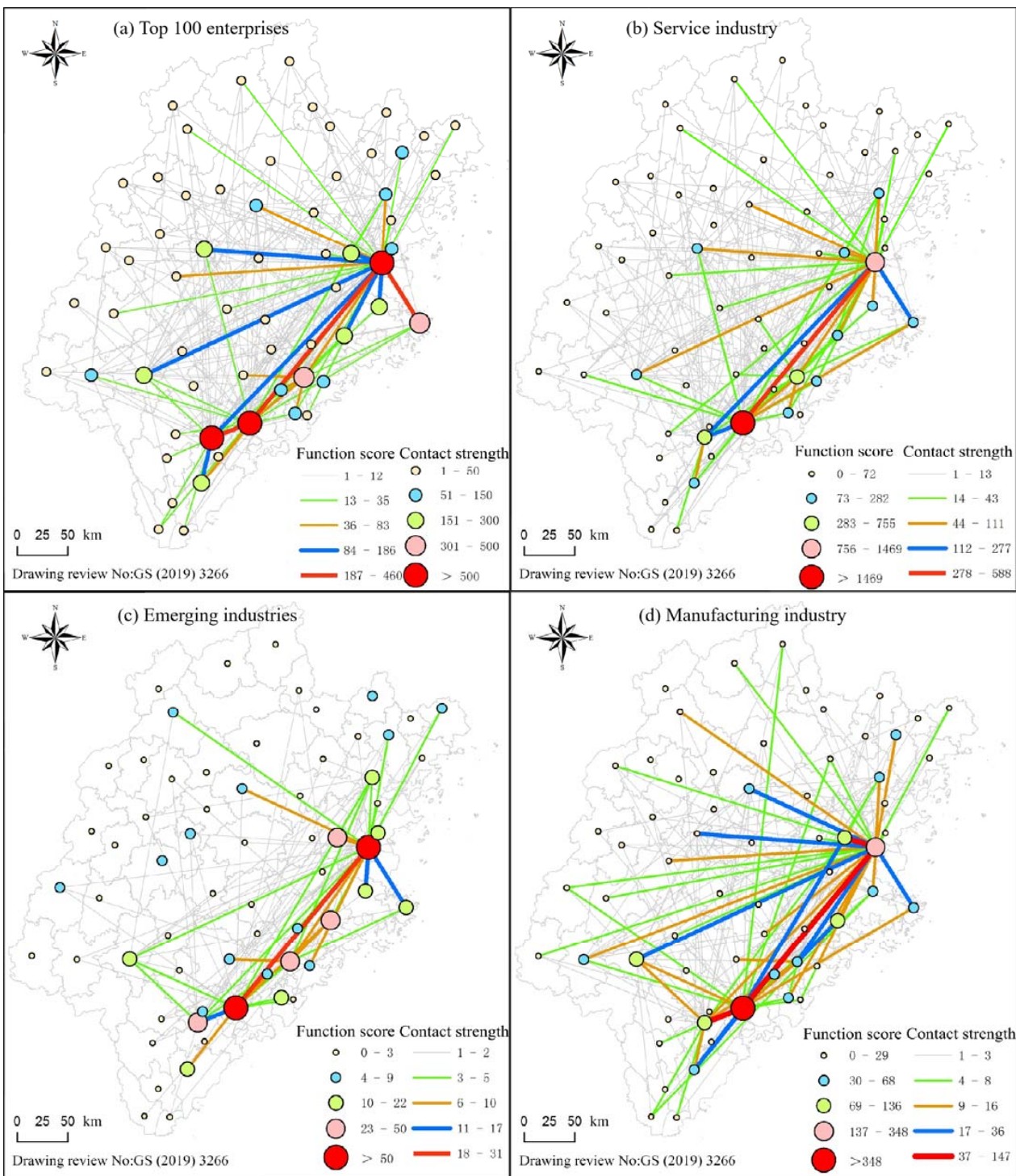

**Figure 3.** County function scores and correlation strength in Fujian Province.

### 3.2. Analysis of Factors Affecting Urban Network in Fujian Province

This study further analyzes the sensitivity of different enterprise networks to various factors. According to the results of the Geo Detector analysis, nine factors of the local economic scale, local key resources and scale and institutional modules influence the spatial distribution pattern of urban network centrality, and the explanatory power of each influencing factor is in the range of 13% to 82%; most of the *p*-values are at the level of $p \leq 0.01$. The correlations of the four groups of data differ greatly from each factor of the detection module, indicating that the urban network formation process is the result

of the joint action and interaction of place, scale and system, and the sensitivity of the different nature of enterprise networks to each factor is different (Table 4). Emerging industries have the strongest correlation with the economic scale and social services, and weaker correlation with indicators of the same metropolitan area, science and technology expenditure and distance to coastal cities. The manufacturing industry, on the other hand, is most influenced by the public services, administrative level and development zone level factors, but is hardly influenced by the indicators of the same metropolitan area and science and technology expenditure. The service industry is most influenced by the factors of the administrative level and social service and development zone level, and almost insensitive to the indicators of the same synergy area. The top 100 enterprises are sensitive to all factors except the same synergy area factor. The gross local product, resident population, number of hospital beds, administrative level and development zone level are all sensitive to the same synergy area factor. The zone level has significant effects on the different nature of enterprises. The effects of the gross local product, resident population, number of hospital beds, administrative level and development zone level are significant for different types of enterprises.

**Table 4.** Analysis results of influence factors of different enterprises in Fujian Province.

| Detection Module | Sequence | Detection Factor | Top 100 Enterprises | | Manufacturing | | Emerging Industry | | Service Industry | |
|---|---|---|---|---|---|---|---|---|---|---|
| | | | Correlation | *p*-Value | Correlation | *p*-Value | Correlation | *p*-Value | Correlation | *p*-Value |
| Local economic scale | X1 | Gross local product | 0.624 | 0.000 | 0.700 | 0.000 | 0.656 | 0.000 | 0.806 | 0.000 |
| | X2 | Resident population | 0.592 | 0.000 | 0.576 | 0.000 | 0.641 | 0.000 | 0.722 | 0.000 |
| Local key resources | X3 | Number of hospital beds | 0.665 | 0.000 | 0.798 | 0.000 | 0.710 | 0.000 | 0.704 | 0.000 |
| | X4 | Science and technology expenditure | 0.568 | 0.025 | 0.048 | 0.652 | 0.573 | 0.016 | 0.104 | 0.423 |
| | X5 | Administrative level | 0.647 | 0.000 | 0.779 | 0.000 | 0.703 | 0.000 | 0.665 | 0.000 |
| Scale | X6 | The same metropolitan area | 0.098 | 0.089 | 0.000 | 0.871 | 0.186 | 0.083 | 0.013 | 0.502 |
| | X7 | The same synergy area | 0.002 | 0.746 | 0.489 | 0.157 | 0.000 | 0.963 | 0.507 | 0.225 |
| Policy | X8 | Distance to coastal cities | 0.260 | 0.002 | 0.139 | 0.106 | 0.208 | 0.011 | 0.178 | 0.119 |
| | X9 | Development zone level | 0.678 | 0.000 | 0.820 | 0.000 | 0.751 | 0.000 | 0.727 | 0.000 |

Note: $p < 0.1$ indicates that there is a 90% probability that the experimental results are significant.

## 4. Discussion

With the development of the division of labor in the product value chain and the specialization of city functions, the network link structure model between cities is being reshaped. This study quantifies the network framework at the local scale using a corporate network perspective, and improves the research and related theories of the urban network structure in the context of scale, place and policy.

Local factors are unique and sticky factors in the layout of enterprises and the formation of urban networks. The local economic scale, social services and level of public service facilities are important factors in the location selection of manufacturing, service and emerging industries, which further affects the formation of urban networks in Fujian Province. Among them, emerging industries have the strongest correlation with the economic scale and social services, while the manufacturing industry is most influenced by public services and the service industry is most influenced by science and technology expenditure. However, manufacturing and emerging industries are not significant for the indicator of science and technology expenditure. Different cities will attract different business value chain blocks and links, which in turn affects the hierarchical differentiation of the city network centrality status and the strength of inter-city linkage relationships, and also leads to a functional division of the urban economy, where the development gap between cities in the flow space tends to widen further [14,20].

Compared with local factors, scale factors play a more complicated role, and the multi-scale nested urban network formed by cities with different administrative scales and governance scales plays an important role in regional economic development. The power division with the administrative level as the scale has an impact on all four groups of enterprise data, indicating that the regional level can effectively promote the development of enterprises. The higher the regional level, the richer the political, economic, talent, tech-

nology and other resources it has, the more it has a voice in regional economic development and the more it can attract enterprises. The top 100 enterprises and service enterprises are greatly influenced by the governance scale of the Xiamen, Zhangquan and Fuzhou metropolitan area, but the governance scale of northeast and southwest Fujian collaborative areas implemented by Fujian Province is not sensitive to the four sets of data, and the policy implementation needs to be verified. The main reason is that the location layout of enterprises is affected by market rules, which limits the role of the government-guided collaborative area layout. For example, the hinterland of the Fuzhou urban network has broken the border of northeast Fujian and expanded to most areas of southwest Fujian in manufacturing.

Policy is the key institutional mechanism factor for the formation of the urban network, which is related to China's gradualism and hierarchical experimental governance model, mainly reflected in two aspects: the system of special policy areas in development zones and the changes of regional development policies (regional development strategies). Since the reforming and opening up, southeastern Fujian Province has become a key area for national comprehensive development, and the province's regional development policies have gone through the process of balanced development from coastal Xiamen, Zhangquan and Fuzhou areas to western mountainous areas, and from coastal Xiamen and Fuzhou cities to coastal counties and cities to western mountainous areas. The closer it is to the coastal areas, the more favorable it is to make use of the relevant policy advantages of the state and Fujian Province, and the more favorable it is to the development layout of enterprises. There is a significant correlation between the top 100 enterprises and the service industry, and it also has a great influence on the urban linkage. The special economic policies of development zones, which have been implemented for a long time, are the spatial carriers for the concentration of various types of enterprises and policy regulation, as well as the multi-scale platform for the connection of urban enterprise networks, and have an important impact on all four groups of data. The manufacturing industry relies on the development zone system most strongly.

At present, in terms of an urban analysis, research has been performed from the perspective of global cities to analyze the problems of urban gentrification and capital concentration in the process of capitalist urban revival [37–39], from the perspective of urban design and public space transformation to guide the transformation of the city to a healthy city [40–42], from a historical perspective to explore the issue of infrastructure construction guiding the future of the city [43] and from the background of urban sprawl to discuss the issue of brownfield/urban open space mixed use [44,45]. However, there is still a lack of research on the characteristics of the network between cities and the governance of the flow space in general. Therefore, the contribution of this paper is the analysis of the network characteristics and governance problems of regional urban development in the flow space, and establishment of a research framework of the urban network relationship under the coordination of scale, place and policy, which provides scientific evidence for the formulation of provincial development policies. On the one hand, the centrality and network status of urban development have broken the restriction of the "administrative regional economy", and it is urgent for the government to establish governance scales that are dynamically configured with enterprise networks, such as metropolitan areas (circles). On the other hand, the overall planning of land space in each province needs the spatial pattern of networked towns, and the multi-scale nesting model, especially the overall function of the coastal urban network belt, should be fully considered in policy making to avoid the new phenomenon of the "administrative region economy" caused by the separation of collaborative areas. Finally, in the production division environment, the specialization division of the urban network formed along the functional division of the value chain is a remarkable feature of the economic landscape of the regional urban system, and small- and medium-sized cities can fully occupy a place in the urban network because of key resources. Therefore, the regional policy for small- and medium-sized cities must become a policy that activates the local and leverages the global, regroups local and

non-local assets, leverages key resources and creates advantages in the functional division of labor in specialized cities.

This study explores the relationship between different types of enterprises and influencing factors, which provides key resources for different types of enterprises to recombine local and non-local assets. According to the spatial directivity and polarization effect of manufacturing, service and emerging industries, we can formulate corresponding local governance policies to guide and better serve enterprises of different natures and provide a decision-making basis for the functional division of specialized cities. This is also a way to promote the expansion of the "flow space" urban network theory to network governance applications.

Although this study has contributed to the study of local spatial governance, it still has some limitations. First, it only analyzes the data of urban network characteristics in Fujian Province in 2021, and lacks the analysis of the evolution dynamics of urban networks; second, it lacks the attention to the edge cities in mountainous areas; finally, the paper only analyzes the urban network linkage of the top 100 enterprises in Fujian Province, and lacks the analysis of network characteristics from a national–local perspective.

## 5. Conclusions

This paper identifies the characteristics of network heterogeneity at the provincial scale in Fujian Province through the quantitative analysis of different types of enterprise networks using the scale, place and policy synergy framework as a theoretical perspective. The main findings can be summarized as follows.

First, the network linkage presents overall strong coastal centripetal and small-world network characteristics, and the urban network linkage reflected by different types of enterprises has obvious spatial directionality and polarization effects, presenting a near-regional dense network and non-geographic proximity to the network hinterland of the economic geography landscape.

Second, coastal cities have strong centrality, and the specialized division of urban functions emerges. The large cities of Fuzhou, Xiamen, Zhangzhou and Quanzhou in the city system become the agglomeration of headquarters of different types of enterprises, while the small- and medium-sized cities of Zhangpu, Fuqing and Minhou carry a large number of processing and assembly enterprises.

Third, the network links present multidimensional nested schema, and the network hierarchy and network linkage patterns reflected by different industries vary significantly. For the top 100 enterprises, the urban network in Fujian Province shows a multicenter, flat and orderly hierarchical structure, and the linkage shows a multidimensional nested schema featuring a dense network of coastal cities, axis of mountain–sea linkage and two major coastal metropolitan areas. The service and manufacturing industries have a hierarchical pyramidal urban structure with Xiamen as a single center and a multidimensional nested schema with core circles, axes and networks. The emerging industries form a network of cities with Xiamen and Fuzhou as the twin centers, showing a coastal belt pattern and a radial linkage pattern with Xiamen and Fuzhou as the central circles.

Fourth, during the formation of the urban network, emerging industries have the strongest correlation with the economic scale and social services, while manufacturing is most influenced by the public services, administrative level and development zone level. The service industry, on the other hand, is most influenced by science and technology expenditure, and the same metropolitan area. The top 100 enterprises are sensitive to all factors except for the same synergy zone factor. The distance from the coast has a significant effect on all types of enterprises.

In urban network governance, the relationship of urban networks under the synergy of scale, place and policy is dynamic. This paper is only a study of urban networks based on enterprise networks, which has limitations for grasping the relationship between urban networks as a whole. Urban network governance involves spatial development, economic

equilibrium and changes in the subject of governance, and there is still a long way to go to improve local spatial governance.

**Author Contributions:** J.Z.: Conceptualization, Methodology, Software, Data curation, Writing—Original draft preparation, Visualization, Writing—Reviewing and Editing. S.W.: Conceptualization, Methodology, Writing—Reviewing, Visualization, Data curation. Q.S.: Conceptualization, Methodology, Writing—Reviewing, Writing—Reviewing and Editing. All authors have read and agreed to the published version of the manuscript.

**Funding:** The National Natural Science Foundation of China (41901159); Fujian Science and Technology Plan Project (2021R0107); and Fujian Natural Science Foundation of China (2022J0113). Social Science Planning Project of Fujian Province (No. FJ2020C036); The Projects of Fujian Social Science Foundation (No. FJ2022C097).

**Institutional Review Board Statement:** Not applicable.

**Informed Consent Statement:** Informed consent was obtained from all subjects involved in this study.

**Data Availability Statement:** The datasets used and/or analyzed during the current study are available from the corresponding author on reasonable request.

**Conflicts of Interest:** The authors declare no conflict of interest.

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
