# Peer review of "Research on the Characteristics and Influencing Factors of Provincial Urban Network from the Perspective of Local Governance—Based on the Data of the Top 100 Enterprises in Four Categories in Fujian Province"

_sustainability, doi:10.3390/su15129368_

Round 1

Reviewer 1 Report

While this manuscript is already quite well written, I would recommend some further improvements and clarifications and/or (if space/ word count allows) some elaborations in the Introduction, Methodology and Results sections. Some of these clarifications and elaborations may make your article a bit easier to understand for an international reading audience; some of the concepts you use may be well-known in China but much less so in other countries?

1. Introduction:

- Because (as is usual in this journal) this is in fact introduction and literature review/ theoretical framework in one section, a lot of concepts and  approaches are briefly mentioned, but not really discussed more in-depth. If the limited space/ word count available allows I would recommend discussing your key concepts in a bit more detail. This may also make clearer what your conceptual framework is, what is in your view still missing in the debate on urban networks so far and what you intend to add to this debate.

- Being an urban geographer myself, I should ask you: are you really discussing and analysing urban networks, or rather corporate networks/ networks of companies? Can you try to add/ clarify where 'the urban' is in your analysis?

- Scale and terminology (also in the next sections) are sometimes a bit confusing, mixing 'urban' and 'provincial'?

2. Methodology:

- 2.1 Research cases and data sources: add an argumentation why you chose Fujian Province as your case study area. Why is Fujian a good choice as a case study of what you want to analyse?

- Also in 2.1 Research cases and data sources: explain what 'emerging enterprises' are? Why and how are they 'emerging'? To which economic sectors do these enterprises belong?

- 2.2 Research framework: what is Geo Detector?

- Figure 2 and Table 1: how/ why is 'distance to coastal cities' a policy factor?

- Also in Figure 2 and Table 1: why do you use gross national product here? Shoudln't it be gross local product or gross regional product?

3. Result analysis:

- 3.1.1: What are 'small-world network characteristics'?

- 3.1.2, page 9: "Thirdly, the in-degree and out-degree of the urban network show a correlation...". Correlation? Did you calculate this? It is not in Table 3?

Author Response

Response to Reviewer

We have greatly appreciated the reviewers’ efforts. Comments and feedback were very constructive and able to improve the quality of the manuscript.

My Manuscript: sustainability-2351132

Article titled: "Research on the characteristics and influencing factors of provincial urban network from the perspective of local governance——Based on the data of the top 100 enterprises in four categories in Fujian Province"

The following is a summary of the reviewers' comments.

Reviewer 1:

Suggestions:1. Because (as is usual in this journal) this is in fact introduction and literature review/ theoretical framework in one section, a lot of concepts and  approaches are briefly mentioned, but not really discussed more in-depth. If the limited space/ word count available allows I would recommend discussing your key concepts in a bit more detail. This may also make clearer what your conceptual framework is, what is in your view still missing in the debate on urban networks so far and what you intend to add to this debate.

Author reply: Thanks to the reviewer for the reminder. We added descriptions of related concepts. And added the purpose and contribution of our city network. Such as line number 49-54, 99-118, 201-205, 502-529, 304-307, 139-147.

Line 201-205 “Network density refers to the closeness of the relationship between nodes in the network, which expresses the closeness of the connection between nodes. Centrality reflects the status of nodes in the network. Network centralization is an analysis of network integration based on centrality, which shows the overall integration of the network.”

Line 304-307 “The main feature of small-world network is that it has a relatively small average path length and a relatively large clustering coefficient, that is, the clustering of the three major industries is high, but the systems are relatively independent.”

Line 139-147 “The manufacturing industry in this study refers to the industry in the national economy that uses certain resources (materials, energy, equipment, etc.) to transform them into products that can be used and utilized by people through the process of mechanical manufacturing. The service industry refers to the collection of production departments and enterprises engaged in service products. Emerging industries refer to industries that are based on major technological breakthroughs and major development needs, have a major leading role in the overall economic and social development and long-term development, and have huge growth potential.”

Line 49-54“Urban networks in China are multi-scale and complex, covering a large geographical area with significant regional differences. Under the role of different geographical environments and the coordination of scale, place and policy[7], the patterns of urban network relationships and influencing factors become diverse[8]. As a result, there is still a need for a breakthrough in the analysis of urban networks and the refinement of related theories in the context of scale, place and policy. ”

Line 99-118“As we can see above, the research on urban networks based on enterprise networks has yielded rich results, but at the same time, the academic community has become increasingly aware of the fact that the research on existing urban networks has neglected the influence of industry development characteristics and institutional environment[26, 30], ignored the research on local regional scale such as provincial areas[31], and lacked dialogue with local governance policies and insufficient policy evaluation. Therefore, this paper attempts to address the above issues by integrating the post-disciplinary perspectives of relational economic geography, evolutionary economic geography and political economic geography as well as current spatial governance theories. Using the research framework of the coordination of scale, place and policy, this paper takes enterprises in different industries in Fujian Province as the empirical scale and object to explore the characteristics and influencing factors of the provincial urban network patterns corresponding to different enterprise networks. The main objectives are, firstly, to establish a research framework of urban network relations under the coordination of scale, place and policy, which can be used to explore the networked characteristics and governance issues of provincial urban development under the flow space; secondly, to explore the relationship between different types of enterprises and the influencing factors, and to regroup local and non-local assets for different types of enterprises to take advantage of key resources and form a specialized urban functional division of labor.”

Line 502-529 “At present, in terms of urban analysis, from the perspective of global cities to analyze the problems of urban gentrification and capital concentration in the process of capitalist urban revival[37-39], from the perspective of urban design and public space transformation to guide the transformation of the city to a healthy city[40-42], from a historical perspective to explore the issue of infrastructure construction guiding the future of the city[43], from the background of urban sprawl to discuss the issue of brownfield/urban open space mixed use[44, 45]. However, there is still a lack of research on the characteristics of the network between cities and the governance of the flow space in general. Therefore, the contribution of this paper is to analyze the network characteristics and governance problems of regional urban development in the flow space, and establish a research framework of urban network relationship under the coordination of scale, place and policy, which provides scientific evidence for the formulation of provincial development policies. On the one hand, the centrality and network status of urban development have broken the restriction of "administrative regional economy", and it is urgent for the government to establish governance scales that are dynamically configured with enterprise networks, such as metropolitan areas (circles). On the other hand, the overall planning of land space in each province needs the spatial pattern of networked towns, and the multi-scale nesting model, especially the overall function of coastal urban network belt, should be fully considered in policy making to avoid the new phenomenon of "administrative region economy" caused by the separation of collaborative areas. Finally, in the production division environment, the specialization division of urban network formed along the functional division of value chain is a remarkable feature of the economic landscape of regional urban system, and small and medium-sized cities can fully occupy a place in the urban network because of key resources. Therefore, the regional policy for small and medium-sized cities must become a policy that activates the local and leverages the global, regroups local and non-local assets, leverages key resources, and creates advantages in the functional division of labor in specialized cities.”

Suggestions:2. Being an urban geographer myself, I should ask you: are you really discussing and analysing urban networks, or rather corporate networks/ networks of companies? Can you try to add/ clarify where 'the urban' is in your analysis?

Author reply: Thanks to the reviewer for your suggestion. We added the role of cities and the role of firm networks in the provincial network analysis. Such as line 80-82, 70-74.

Line 80-82 “This study explores the urban network relationship of enterprises from the perspective of corporate network, and can quantify its urban network mechanism from specific things. ”

Line 70-74 “As the basic unit of a province, a city is the basic component of analyzing the internal network of a province. By quantifying the network relationship of each city in this study, the network relationship between cities can be grasped from the provincial level as a whole.”

Suggestions:3. cale and terminology (also in the next sections) are sometimes a bit confusing, mixing 'urban' and 'provincial'?

Author reply: Thanks to the reviewer for your suggestion. We added city and province domain relations. Such as line 70-74.

Line 70-74 “As the basic unit of a province, a city is the basic component of analyzing the internal network of a province. By quantifying the network relationship of each city in this study, the network relationship between cities can be grasped from the provincial level as a whole.”

Suggestions:4. 2.1 Research cases and data sources: add an argumentation why you chose Fujian Province as your case study area. Why is Fujian a good choice as a case study of what you want to analyse?

Author reply: Thanks to the reviewer for the reminder. We have added the reason for choosing Fujian Province as the case study area. Such as line 121-133.

Line 121-133 “This study takes Fujian Province as a case study, the main starting points are: 1. Fujian Province is located on the southeast coast of China, and now governs nine cities and one district (Pingtan Comprehensive Experimental Zone). The geographical units of mountains and seas in Fujian Province differ significantly. It was an important pilot area for the advancement of industrialization and market-oriented reforms in early China. 2. The focus of the province's regional development policy has gone through a process from the coastal Xiamen-Zhangzhou-Quanzhou region to the Fuzhou region and to the western mountainous regions. At present, metropolitan circles of different sizes and administrative levels have been formed. 3. At the same time, Fujian Province is a province with frequent interactions between central and local levels of urban power, and the policy governance process is continuous, typical and representative. 4. There are more than 1.32 million enterprises with a registered capital of more than one million, which have a data foundation for network analysis.”

Suggestions:5. Also in 2.1 Research cases and data sources: explain what 'emerging enterprises' are? Why and how are they 'emerging'? To which economic sectors do these enterprises belong?

Author reply: Thanks to the reviewer for the reminder. We have added explanations for manufacturing, services, and emerging industries. Line 139-147.

Line 139-147 “The manufacturing industry in this study refers to the industry in the national economy that uses certain resources (materials, energy, equipment, etc.) to transform them into products that can be used and utilized by people through the process of mechanical manufacturing. The service industry refers to the collection of production departments and enterprises engaged in service products. Emerging industries refer to industries that are based on major technological breakthroughs and major development needs, have a major leading role in the overall economic and social development and long-term development, and have huge growth potential.” 

Suggestions:6. 2.2 Research framework: what is Geo Detector?

Author reply: Thanks to the reviewer for the reminder. We have added relevant explanations. Such as lines 230-236.

line 230-236 “Geo Detector is a set of statistical methods for detecting spatial variability and revealing the driving forces behind it. Its core idea is based on the assumption that if an independent variable has an important influence on a dependent variable, then the spatial distribution of the independent variable and the dependent variable should be similar. This paper uses the Factor Detection module in Geo Detector to study the magnitude and variation in the degree of explanation of each influencing factor in the centrality of urban enterprise networks in Fujian.”

Suggestions:7. Figure 2 and Table 1: how/ why is 'distance to coastal cities' a policy factor?

Author reply: Thanks to the reviewer for the reminder. We added why "distance from coastal cities" is a policy factor. Such as line 278-280.

Line 278-280 “At the policy level, the policy of coastal priority development is conducive to enterprise network development[35]. Fujian Province implements the development policy of coastal cities driving inland cities.”

Suggestions:8. Also in Figure 2 and Table 1: why do you use gross national product here? Shoudln't it be gross local product or gross regional product?

Author reply: Thanks to the reviewer for the reminder. We changed all “gross natural product” to “gross local product”. This is a more precise wording.

Suggestions:9. 3.1.1: What are 'small-world network characteristics'?

Author reply: Thanks to the reviewer for the reminder. We have added corresponding explanations. Such as lines 304-307.

line 304-307 “The main feature of small-world network is that it has a relatively small average path length and a relatively large clustering coefficient, that is, the clustering of the three major industries is high, but the systems are relatively independent.”

Suggestions:10. 3.1.2, page 9: "Thirdly, the in-degree and out-degree of the urban network show a correlation...". Correlation? Did you calculate this? It is not in Table 3?

Author reply: Thanks to the reviewer for the reminder. This conclusion is similar to Table 3, therefore, we deleted the corresponding statement.

Reviewer 2 Report

The research is quite interesting and has been carried out carefully (methods, figures, tables, etc.) but it lacks of a literature review and there is also the need to provide a connection between the literature review (which can be an added paragraph) and the discussion.

I highly suggest to put emphasis on a number of elements in the literature review apart from the main themes of the paper:

Fist, the distortions provoked by capitalist urban regeneration processes by taking into references some examples of global cities:

- 2020. Alpha City: How London Was Captured by the Super-Rich. London: Verso

- 2019. From “Ribera Plan” to “Diagonal Mar”, passing through 1992 “Vila Olímpica”. How urban renewal took place as urban regeneration in Poblenou district (Barcelona). Land Use Policy, 89, 10422

- 2019. Capital City. Gentrification and the real estate state. London-New York: Verso

2017. The icon project: architecture, cities and capitalist globalization. New York: Oxford University Press

Second, on healthy cities and urban regeneration solutions, also taking into account the change of urbanisation paradigms after the 2020 pandemic outbreak. See the following references:

- (2020). New Healthy Settlements Responding to Pandemic Outbreaks: Approaches from (and for) the Global City. The Plan Journal, 5(2): 385-406

- 2020. Changing the urban design of cities for health: The superblock model. Environment International, 134: 105132

2020). The impact of COVID-19 on public space: an early review of the emerging questions – design, perceptions and inequities. Cities & Health

Third, On infrastructure construction, give an international overview of the literature released on this aspects also wiht an historical point of view, such as:

Vitiello, D. (2017) Planning for infrastructure. Lifelines, mobility and urban development. In The Routledge handbook of planning history (pp. 325–337). Routledge

(2010). City planning and infrastructure: Once and future partners. Journal of Planning History9(1), 21–42

2001). Splintering Urbanism. Networked Infrastructures, Technological Mobilities and the Urban Condition. Routledge

(1985). Infrastructure and urban growth in the Nineteenth Century. Chicago: Public Work Historical Society

Fourth, in a context in which urban sprawl is taking place, what to do with brownfields/urban voids? Please, mention it by taking into account these references:

- 2022. Brownfield infrastructures. In: The Elgar Companion to Urban Infrastructure Governance, open access here https://uvadoc.uva.es/handle/10324/58636

- 2020, A Glossary of Urban Voids, JOVIS Verlag, Berlin

Please, connect the literature review with the discussion.

Highlight better the lessons learned and the future lines of research starting from this interesting case study.

I expect a new version of the paper to review.

Author Response

Response to Reviewer

We have greatly appreciated the reviewers’ efforts. Comments and feedback were very constructive and able to improve the quality of the manuscript.

My Manuscript: sustainability-2351132

Article titled: "Research on the characteristics and influencing factors of provincial urban network from the perspective of local governance——Based on the data of the top 100 enterprises in four categories in Fujian Province"

The following is a summary of the reviewers' comments.

Reviewer 2:

Suggestions:1. The research is quite interesting and has been carried out carefully (methods, figures, tables, etc.) but it lacks of a literature review and there is also the need to provide a connection between the literature review (which can be an added paragraph) and the discussion.

I highly suggest to put emphasis on a number of elements in the literature review apart from the main themes of the paper:

Fist, the distortions provoked by capitalist urban regeneration processes by taking into references some examples of global cities:

- 2020. Alpha City: How London Was Captured by the Super-Rich. London: Verso

- 2019. From “Ribera Plan” to “Diagonal Mar”, passing through 1992 “Vila Olímpica”. How urban renewal took place as urban regeneration in Poblenou district (Barcelona). Land Use Policy, 89, 10422

- 2019. Capital City. Gentrification and the real estate state. London-New York: Verso

2017. The icon project: architecture, cities and capitalist globalization. New York: Oxford University Press

Second, on healthy cities and urban regeneration solutions, also taking into account the change of urbanisation paradigms after the 2020 pandemic outbreak. See the following references:

- (2020). New Healthy Settlements Responding to Pandemic Outbreaks: Approaches from (and for) the Global City. The Plan Journal, 5(2): 385-406

- 2020. Changing the urban design of cities for health: The superblock model. Environment International, 134: 105132

2020). The impact of COVID-19 on public space: an early review of the emerging questions – design, perceptions and inequities. Cities & Health

Third, On infrastructure construction, give an international overview of the literature released on this aspects also wiht an historical point of view, such as:

Vitiello, D. (2017) Planning for infrastructure. Lifelines, mobility and urban development. In The Routledge handbook of planning history (pp. 325–337). Routledge

(2010). City planning and infrastructure: Once and future partners. Journal of Planning History9(1), 21–42

2001). Splintering Urbanism. Networked Infrastructures, Technological Mobilities and the Urban Condition. Routledge

(1985). Infrastructure and urban growth in the Nineteenth Century. Chicago: Public Work Historical Society

Fourth, in a context in which urban sprawl is taking place, what to do with brownfields/urban voids? Please, mention it by taking into account these references:

- 2022. Brownfield infrastructures. In: The Elgar Companion to Urban Infrastructure Governance, open access here https://uvadoc.uva.es/handle/10324/58636

- 2020, A Glossary of Urban Voids, JOVIS Verlag, Berlin

Please, connect the literature review with the discussion.

Author reply: We thank the reviewer for such a comprehensive literature, which we have added to the Discussion section to highlight the depth and reach of the article. Such as line 502-529.

line 502-529 “At present, in terms of urban analysis, from the perspective of global cities to analyze the problems of urban gentrification and capital concentration in the process of capitalist urban revival[37-39], from the perspective of urban design and public space transformation to guide the transformation of the city to a healthy city[40-42], from a historical perspective to explore the issue of infrastructure construction guiding the future of the city[43], from the background of urban sprawl to discuss the issue of brownfield/urban open space mixed use[44, 45]. However, there is still a lack of research on the characteristics of the network between cities and the governance of the flow space in general. Therefore, the contribution of this paper is to analyze the network characteristics and governance problems of regional urban development in the flow space, and establish a research framework of urban network relationship under the coordination of scale, place and policy, which provides scientific evidence for the formulation of provincial development policies. On the one hand, the centrality and network status of urban development have broken the restriction of "administrative regional economy", and it is urgent for the government to establish governance scales that are dynamically configured with enterprise networks, such as metropolitan areas (circles). On the other hand, the overall planning of land space in each province needs the spatial pattern of networked towns, and the multi-scale nesting model, especially the overall function of coastal urban network belt, should be fully considered in policy making to avoid the new phenomenon of "administrative region economy" caused by the separation of collaborative areas. Finally, in the production division environment, the specialization division of urban network formed along the functional division of value chain is a remarkable feature of the economic landscape of regional urban system, and small and medium-sized cities can fully occupy a place in the urban network because of key resources. Therefore, the regional policy for small and medium-sized cities must become a policy that activates the local and leverages the global, regroups local and non-local assets, leverages key resources, and creates advantages in the functional division of labor in specialized cities.”

Reviewer 3 Report

The paper aims in quantifying urban network structure by using a Chinese region as a case study. The approach is holistic to an adequate degree since it achieves to incorporate most of the crucial aspects of urban network theory and research. The paper uses maps and diagrams in a constructive way.

The paper should use a more simple wording suitable for a more wide audience not familiar with regional science and regional economics. In this sense, some more basic definitions are possibly useful in the introduction. For example, what is an urban network, how do we define centrality or linkage, small world networking, etc?

The authors should be more clear in describing how the scale, place, and policy scheme derive from the literature.

Figure 2 provides a quite complicated framework for which the paper has limited extent to analyze in detail. This is well understood, although a reader needs to understand better for example how the social network analysis has been conducted without questionnaires. At the same time is not clear how the authors measure the connections. We don't understand if they are measured or calculated somehow.

At the same time, more explanation is needed for the processes of the Geodetector analysis, or the validity of specific calculations for example the contact strength (figure 3).

The discussion and conclusions are informative and adequately structured although they fell into several contradictions. For example it cannon be such a conclusion that the process of urban network formation is the result of the joint action of place, scale, and system since this was the framework already set by the authors at the beginning.

A main structural weakness of this paper is that it deals with too many parameters and processes, and at the same time uses too much regional science jargon that makes it difficult accessible by nonregional scientists. This will need a major revision with rewriting throughout the paper.

Author Response

Response to Reviewer

We have greatly appreciated the reviewers’ efforts. Comments and feedback were very constructive and able to improve the quality of the manuscript.

My Manuscript: sustainability-2351132

Article titled: "Research on the characteristics and influencing factors of provincial urban network from the perspective of local governance——Based on the data of the top 100 enterprises in four categories in Fujian Province"

The following is a summary of the reviewers' comments.

Reviewer 3:

Suggestions:1. The paper aims in quantifying urban network structure by using a Chinese region as a case study. The approach is holistic to an adequate degree since it achieves to incorporate most of the crucial aspects of urban network theory and research. The paper uses maps and diagrams in a constructive way.

Author reply: Thanks to the reviewer for your affirmation of the article, we will make persistent efforts and strive to make some contributions in the discipline.

Suggestions:2. The paper should use a more simple wording suitable for a more wide audience not familiar with regional science and regional economics. In this sense, some more basic definitions are possibly useful in the introduction. For example, what is an urban network, how do we define centrality or linkage, small world networking, etc?

Author reply: Thanks to the reviewer for the reminder. We have added relevant explanations. Lines 201-205, 304-307, 139-147.

Line 201-205 “Network density refers to the closeness of the relationship between nodes in the network, which expresses the closeness of the connection between nodes. Centrality reflects the status of nodes in the network. Network centralization is an analysis of network integration based on centrality, which shows the overall integration of the network.”

Line 304-307 “The main feature of small-world network is that it has a relatively small average path length and a relatively large clustering coefficient, that is, the clustering of the three major industries is high, but the systems are relatively independent.”

Line 139-147 “The manufacturing industry in this study refers to the industry in the national economy that uses certain resources (materials, energy, equipment, etc.) to transform them into products that can be used and utilized by people through the process of mechanical manufacturing. The service industry refers to the collection of production departments and enterprises engaged in service products. Emerging industries refer to industries that are based on major technological breakthroughs and major development needs, have a major leading role in the overall economic and social development and long-term development, and have huge growth potential.”

Suggestions:3. The authors should be more clear in describing how the scale, place, and policy scheme derive from the literature.

Author reply: Thanks to the reviewer for the reminder. We have added a corresponding explanation and supplemented Section 2.4. Line 246-286.

Line 246-286 “The scale, place and policy synergy framework perspective proposed by Gong[7] provides an integrated analytical thinking of scale, place and policy synergy for understanding the evolution of regional economic development . It deepens the understanding of the political-economic spatial unity of economic network linkages and political governance spatial reconfiguration, enables the analysis of regional economy "beyond borders"[1], and promotes the development of the network paradigm-based relational geography theory to network governance applications. Scale refers to one or several levels of representation, experience, and organization of geographic events and processes. In this research, this study refers to the administrative scale of provinces, cities, and counties, and the governance scale of metropolitan areas. Place refers to a specific area relative to large-scale areas such as the world and a country, and is a city or county area with special natural and socio-economic factors. Policy refers to government-led development planning, spatial development priorities, layout of special economic zones, and other specific socio-economic measures. Among the local factors, city population size and economic scale influence the linkage relationship between cities, and the contribution of different industries in the economic network varies[25]. The level of local public facilities, science and technology expenditure, and social service conditions also have an important impact on the layout and development of enterprises[32, 33], and these key resources become important factors to be considered when selecting the location of enterprise networks. Therefore, this study uses the indicators of gross local product, resident population, number of hospital beds, and science and technology expenditure to quantify the local influencing factors. At the scale level, the preferences of firms' location choices and the network concentration process they lead to are important micro mechanisms of urban relationship patterns[21, 34]. Governments of different administrative levels have different resource disposal capabilities. The higher the administrative level, the more political power, policy advantage and information advantage it has, which is conducive to enhancing the competitiveness of product value chain and directly affects the location choice of company headquarters and branches[34]. The metropolitan area development and synergy area development promoted by the state are conducive to the formation of enterprise networks near the region[31]. Therefore, in this study, the administrative level, the same metropolitan area and the same synergy area indicators are chosen to quantify the influencing factors of the scale. At the policy level, the policy of coastal priority development is conducive to enterprise network development[35]. Fujian Province implements the development policy of coastal cities driving inland cities. Special development zones established by the state build economic links between different spatial scales, and it is conducive to attracting business clusters. The spatial organization of multi-zone enterprises is influenced by the regional development context, development regime and policies[1, 2, 31, 35]. Therefore, two indicators, distance to coastal cities and development zone level, are chosen in this study to quantify the influencing factors of policies.”

Suggestions:4. Figure 2 provides a quite complicated framework for which the paper has limited extent to analyze in detail. This is well understood, although a reader needs to understand better for example how the social network analysis has been conducted without questionnaires. At the same time is not clear how the authors measure the connections. We don't understand if they are measured or calculated somehow. At the same time, more explanation is needed for the processes of the Geodetector analysis, or the validity of specific calculations for example the contact strength (figure 3).

Author reply: Thanks to the reviewer for the reminder. We explained and added their calculation formulas in the Methods section. Such as line 201-244.

line 201-244 “Network density refers to the closeness of the relationship between nodes in the network, which expresses the closeness of the connection between nodes. Centrality reflects the status of nodes in the network. Network centralization is an analysis of network integration based on centrality, which shows the overall integration of the network.

Calculation formula (2) of Network centrality :

(2)

Calculation formula (3) of Network centralization :

(3)

Calculation formula (4) of Network density :

(4)

Where,  is the number of neighbors of the node, N is the set of all nodes in the network, |N| is the number of nodes in the network,  is the maximum value of the network centrality, and L is the connection possessed by the network practice.

(3)City Rank Score

The city rank score Si of a county or city is equal to the product of the number and weight of corporate headquarters established in that city, plus the product of the number and weight of branches established in that city, as shown in formula 5.

(5)

where  refers to the number of headquarters of top 100 enterprises in city i (h for headquarters);  refers to the number of branches in that city (b for branches); and q is the weight (=2, =1).

This study assigns scores of 2 and 1 to the headquarters and branches of enterprises respectively. According to formula (5), the scores of counties and cities in the urban network of Fujian Province are calculated and classified into five levels according to the natural fracture method, and thus the network hierarchy and multidimensional nested schema of cities can be explored.

(4)Geo Detector

Geo Detector is a set of statistical methods for detecting spatial variability and revealing the driving forces behind it. Its core idea is based on the assumption that if an independent variable has an important influence on a dependent variable, then the spatial distribution of the independent variable and the dependent variable should be similar. This paper uses the Factor Detection module in Geo Detector to study the magnitude and variation in the degree of explanation of each influencing factor in the centrality of urban enterprise networks in Fujian. The expressions of the factor detection module are as follows:

(6)

where q is the value of the detection power of the influencing factor on the network centrality; h=1,..., L is the grading of each factor of the variable; is the total variance of the network centrality of cities in Fujian province, and  is the variance of the network centrality of cities in Fujian province; N is the number of cities in Fujian province; Nh is the number of types of the influencing factor X; the value range of q is [0, 1], and the larger the value q, the greater the influence of the factor on Fujian.”

Suggestions:5. The discussion and conclusions are informative and adequately structured although they fell into several contradictions. For example it cannon be such a conclusion that the process of urban network formation is the result of the joint action of place, scale, and system since this was the framework already set by the authors at the beginning.

Author reply: Thanks to the reviewer for your affirmation.

Suggestions:6. “This study constructs a research framework to study the urban network formed by the synergy of scale, place and policy, and adopts Social Network Analysis Methods and Geo Detector to analyze the characteristics and influencing factors of provincial urban networks, by using different types of enterprises in the provincial area as the empirical scale and object. ” not very clear.

Author reply: Thanks to the reviewer for the reminder. We have revised the relevant statements to make them easier to understand. Such as line 14-18.

line 14-18 “This study constructs a research framework to study the urban network formed by the synergy of scale, place and policy. It mainly takes enterprises in different industries in different provinces as the empirical scale and object, and uses methods such as social network and Geo Detector to analyze the characteristics and influencing factors of the provincial network relationship mode of enterprises among cities.”

Suggestions:7. “To sum up, how to study the formation mechanism of urban networks under the coordination of scale, place and policy from the perspective of different industries is a concrete framework to realize the enforceability of local space governance.”How does pattern, externalities and resilience lead as to scale, place and policy. It is a quite arbitrary sum up.

Author reply: Thanks to the reviewer for your suggestion. We added some content to make its summary more convincing. Such as line 67-74, 80-82.

line 67-74“Thirdly, in terms of the spatial scale of research, macroscopic scales involving global, national and regional scales remain the main object of research[213], and research at provincial and sub-provincial scales is still lacking. Thus there is a lack of implementable local spatial governance frameworks. As the basic unit of a province, a city is the basic component of analyzing the internal network of a province. By quantifying the network relationship of each city in this study, the network relationship between cities can be grasped from the provincial level as a whole.”

line 80-82 “This study explores the urban network relationship of enterprises from the perspective of corporate network, and can quantify its urban network mechanism from specific things. ”

Suggestions:8. This is a well aimed comment, well done

Author reply: Thanks to the reviewer for your affirmation.

Suggestions:9. “Fujian Province as the empirical scale ”Why is this area chosen? What is its critical relation with the paper hypotheses?

Author reply: Thanks to the reviewer for your suggestion. We explained it on the case presentation. Such as lines 121-133.

line 121-133 “This study takes Fujian Province as a case study, the main starting points are: (1) Fujian Province is located on the southeast coast of China, and now governs nine cities and one district (Pingtan Comprehensive Experimental Zone). The geographical units of mountains and seas in Fujian Province differ significantly. It was an important pilot area for the advancement of industrialization and market-oriented reforms in early China. (2) The focus of the province's regional development policy has gone through a process from the coastal Xiamen-Zhangzhou-Quanzhou region to the Fuzhou region and to the western mountainous regions. At present, metropolitan circles of different sizes and administrative levels have been formed. (3) At the same time, Fujian Province is a province with frequent interactions between central and local levels of urban power, and the policy governance process is continuous, typical and representative. (4) There are more than 1.32 million enterprises with a registered capital of more than one million, which have a data foundation for network analysis.”

Suggestions:10. “the results of the top 100 enterprises and three major industries showed overall poor connectivity, strong centripetal concentration, incomplete development, and small world characteristics” how this finding came up.

Author reply: Thanks to the reviewer for your suggestion. We've added related instructions. Such as line 301-307.

line 301-307 “Using social network method to measure the related indicators of urban network and According to formula 1-4, we find that the results of the top 100 enterprises and three major industries showed overall poor connectivity, strong centripetal concentration, incomplete development, and small world characteristics (The main feature of small-world network is that it has a relatively small average path length and a relatively large clustering coefficient, that is, the clustering of the three major industries is high, but the systems are relatively independent.). ”

Suggestions:11. “ the process of urban network formation is the result of the joint action of place, scale, and system, and different types of enterprise networks have different sensitivities to each factor.” This was your hypothesis and framework, how can be at the same time the result?

Author reply: Thanks to the reviewer for your suggestion. We have revised the relevant concluding remarks. Such as line 571-577.

line 571-577 “Fourth, During the formation of urban network, emerging industries have the strongest correlation with economic scale and social services, while manufacturing is most influenced by public services, administrative level, and development zone level. The service industry, on the other hand, is most influenced by science and technology expenditure, and the same metropolitan area. The top 100 enterprises are sensitive to all factors except for the same synergy zone factor. The distance from the coast has a significant effect on all types of enterprises.”

Round 2

Reviewer 3 Report

The response to all reviewer's comments was adequate.